# Macronutrient, Micronutrient Supplementation and Monitoring for Patients on GLP-1 Agonists: Can We Learn from Metabolic and Bariatric Surgery?

**DOI:** 10.3390/nu17233659

**Published:** 2025-11-23

**Authors:** Rhea Sibal, G. Balamurugan, Jasmine Langley, Yitka Graham, Kamal Mahawar

**Affiliations:** 1Faculty of Life Sciences and Medicine, King’s College London, London SE1 1UL, UK; rheasibal21@gmail.com; 2Department of Surgery, South Tyneside and Sunderland NHS Foundation Trust, Sunderland SR4 7TP, UK; jasmine.langley3@nhs.net (J.L.); yitka.graham@sunderland.ac.uk (Y.G.); kamal.mahawar@nhs.net (K.M.); 3Helen McArdle Nursing and Care Research Institute, University of Sunderland, Sunderland SR6 0DD, UK; 4Faculty of Biomedical Sciences, Austral University, Buenos Aires B1406, Argentina; 5Faculty of Psychology, University of Anahuac, Mexico City 52786, Mexico

**Keywords:** bariatric surgery, GLP-1 receptor agonists, metabolic and bariatric surgery, micronutrient deficiencies, nutritional monitoring, obesity, protein intake, vitamin and mineral status

## Abstract

**Background/Objectives**: Glucagon-like peptide-1 receptor agonists (GLP-1RAs) are increasingly prescribed for people living with obesity and type 2 diabetes due to their efficacy in reducing appetite and body weight. However, by inducing caloric restriction and altering gastrointestinal physiology, GLP-1RAs may predispose patients to nutritional deficiencies. This review aimed to synthesise current evidence on energy, protein, vitamin, and mineral status in GLP-1RA users, and contextualises these findings with metabolic and bariatric surgery (MBS) guidelines. While metabolic and bariatric surgery (MBS) guidelines mandate structured nutritional monitoring, no equivalent frameworks exist for GLP-1RA therapy, highlighting a critical gap that justifies the need for this review. **Methods**: A narrative review was conducted in three stages: (i) searching PubMed and Embase OVID (August 2025) using MeSH terms and free-text keywords related to GLP-1RAs, micronutrients, and obesity; (ii) screening abstracts and full texts for eligibility; and (iii) synthesising results with comparison to bariatric surgery protocols. Eligible studies included clinical trials, observational cohorts, and reviews reporting nutritional outcomes in GLP-1RA users or describing MBS monitoring guidelines. **Results**: GLP-1RA therapy consistently reduced caloric intake, with frequent inadequacy of protein intake and occasional sarcopenia. Observational data reported that users developed nutritional deficiencies within 12 months, most commonly vitamin D, followed by thiamine and other B vitamins. Mineral deficiencies, particularly in iron, calcium, magnesium, and potassium, were also observed. **Conclusions**: GLP-1RAs are associated with clinically relevant risks of protein, vitamin, and mineral deficiencies. The absence of formal monitoring protocols represents an unmet clinical need, and adaptation of surveillance, as seen in MBS, which may help mitigate long-term complications.

## 1. Introduction

### 1.1. Expanding Use of GLP-1 Receptor Agonists in Obesity Management

Glucagon-like peptide-1 receptor agonists (GLP-1 RAs) have become central to the management of obesity, especially in the current climate with increasing rates of adiposity-related morbidity and mortality globally, supported by a strong evidence base from large cardiovascular outcome trials. Early trials have clearly demonstrated not only safety but also meaningful efficacy, with liraglutide in the LEADER trial achieving significant reductions in cardiovascular risk [1], such as **a multicentre, double-blind outcomes study enrolling 9340 adults with type 2 diabetes and high cardiovascular risk, followed for 3.8 years.** Semaglutide in SUSTAIN-6 produced sustained glycaemic and weight benefits [2], **as shown in a 2-year, randomised, placebo-controlled trial of 3297 participants with type 2 diabetes at elevated cardiovascular risk.** Dulaglutide in REWIND confirmed long-term cardiometabolic improvements [3] **based on a 5.4-year outcomes trial including 9901 adults, many without established cardiovascular disease**. More recently, semaglutide in the STEP-1 trial produced mean weight losses of approximately 15% in individuals living with obesity [4] **in a 68-week randomised trial of 1961 adults with overweight or obesity.** The dual GIP/GLP-1 agonist tirzepatide in SURMOUNT-1 achieved reductions exceeding 20% in some participants [5], **as demonstrated in a 72-week, phase 3 trial enrolling 2539 adults with obesity.** These drugs have paved the way for the efficient invention of even more efficacious drugs in the future. The development of successful pharmacotherapy for obesity and its associated diseases is one of the greatest medical achievements of our age. Many of these drugs attempt to mimic the physiological effects of Metabolic and Bariatric Surgery (MBS). It is, therefore, natural that we try to understand if there is more that we can learn from MBS when using pharmacotherapy for obesity and metabolic syndrome. In this review, we seek to explore the nutritional consequences of pharmacotherapy for obesity and attempt to draw lessons from our experience with MBS.

### 1.2. Emerging Nutritional Concerns and Gaps in Clinical Guidance

Appetite suppression, gastrointestinal side effects, and reduced dietary variety seen with GLP-1 RAs can substantially lower energy intake, sometimes to levels insufficient to meet daily requirements for key micronutrients such as iron, calcium, zinc, and vitamins A, D, E, K, B1, B12, and C [6]. These risks are amplified in individuals with obesity, who frequently present with suboptimal baseline diets and pre-existing deficiencies, alongside obesity-related alterations in nutrient absorption and metabolism [7].

Despite these vulnerabilities, there are currently no consensus guidelines for micronutrient monitoring in patients receiving GLP-1 RAs. This contrasts sharply with MBS, where societies such as the British Obesity and Metabolic Society (BOMSS) [8], the American Society for Metabolic and Bariatric Surgery) [9], the European Society for Clinical Nutrition and Metabolism (ESPEN) [10], and the European Association for Endoscopic Surgery (EAES) [11] provide detailed protocols for baseline screening, ongoing laboratory monitoring, and prophylactic micronutrient supplementation. The lack of guidance in pharmacological obesity care, therefore, highlights a critical gap and provides the rationale for exploring whether lessons from bariatric surgery can inform clinical practice in GLP-1 RAs users.

### 1.3. Physiological Comparisons Between GLP-1 RA and Metabolic and Bariatric Surgery (MBS)

Although pharmacological and surgical approaches differ fundamentally, both GLP-1 receptor agonists and MBS induce weight loss through reductions in appetite, altered nutrient handling, and changes in gastrointestinal physiology [12]. GLP-1 RAs act pharmacologically to slow gastric emptying, suppress appetite, and reduce overall caloric intake, often accompanied by gastrointestinal side effects such as nausea, vomiting, or diarrhoea, which can further limit the variety of food intake in patients [13]. Most of these effects are similar to what is observed after MBS—the only difference being that the bypass of parts of the gastrointestinal tract in some procedures, such as gastric bypasses, can further exacerbate these nutritional issues in particular cases of MBS by affecting the absorption of nutrients [14].

Despite these differences, there is significant overlap in nutritional consequences. Both interventions increase the risk of nutritional deficiencies due to reduced intake, altered absorption, or changes in gastrointestinal physiology [15]. Importantly, while MBS has long been recognised as necessitating structured micronutrient monitoring and supplementation, similar protocols for GLP-1 therapy have not yet been established. Hence, we explore the value of drawing on MBS guidelines to inform the nutritional management of patients on GLP-1 RAs.

## 2. Materials and Methods

A narrative review approach was chosen for its ability to provide a meaningful synthesis of published studies where the subject under investigation is extensive and needs to capture the complexity and interpretation, for which a systematic review would not allow.

This narrative review was conducted in three stages: (i) searching the literature; (ii) screening of abstracts and full texts for relevance; and (iii) synthesis of findings with comparison to bariatric surgery protocols. A comprehensive search was performed in August 2025 across PubMed and Embase OVID to identify relevant publications. The last of these searches was conducted on 27th August. Reference lists of key reviews and guidelines were also hand-searched to capture additional studies. Only English-language publications were included.

Search terms combined MeSH terms and free-text keywords related to GLP-1 receptor agonists and micronutrient outcomes. The following search framework was applied:

**Drug terms:** “Glucagon-Like Peptide 1 Receptor Agonists” [MeSH], “GLP-1 receptor agonist*”, semaglutide, liraglutide, dulaglutide, exenatide, lixisenatide, albiglutide, tirzepatide, “GIP/GLP-1 dual agonist*”.

**Nutrient terms:** “Micronutrients” [Mesh], micronutrient*, vitamin*, mineral*, “nutritional support”, “nutrient deficiency”, “nutrition therapy”, “nutrition assessment”, and “sarcopenia”, “malabsorption”, and “hypoabsorption”.

**Population terms:** “Obesity” [MeSH], obesity, overweight, “weight loss”, “anti-obesity agent*”, “bariatric surgery”, “metabolic surgery”.

Boolean operators (AND/OR) were applied to ensure sensitivity and precision. After removal of duplicates, abstracts were screened for eligibility. Full texts were included if they (i) reported on nutritional or micronutrient outcomes in patients receiving GLP-1 receptor agonists or dual agonists, or (ii) described MBS micronutrient monitoring guidelines relevant to obesity care. Exclusion criteria were studies without nutritional outcomes, animal-only studies, non-English studies or those not reporting on GLP-1 RAs or bariatric populations.

Eligible sources encompassed clinical trials, observational studies, practice guidelines, and narrative or systematic reviews. Bariatric surgery nutrition guidelines from professional bodies (e.g., BOMSS, ASMBS and ESPEN) were also included to provide a comparative framework. Extracted data included reported micronutrient deficiencies, proposed monitoring intervals, and supplementation strategies. A simplified flow diagram summarising the identification, screening, eligibility assessment, and inclusion of studies is presented in Figure 1.

## 3. Results

### 3.1. Impact of GLP-1 Therapy on Energy and Protein Intake

Across the included studies, the type and validity of methods used to characterise nutrient intake and deficiency varied substantially. Dietary intake studies most commonly relied on self-reported 24 h recalls, short food diaries, or appetite questionnaires (e.g., in Christensen et al. [16] and Johnson et al. [17]), all of which are vulnerable to under-reporting in individuals with overweight or obesity. In contrast, higher-quality trials such as Silver et al. [18] and Richardson et al. [19] used weighed 3–7-day food diaries with dietitian-verified nutrient analysis, providing a more reliable measure of energy and protein intake. Biochemical definitions of deficiency were used only in retrospective cohorts such as Butsch et al. [20], where serum B12, ferritin, thiamine, vitamin D, and anaemia levels were extracted from electronic health records. These methodological differences are important when interpreting results, as studies relying solely on reported intake may underestimate true nutrient insufficiency.

GLP-1RA therapy consistently reduces overall energy intake [16,21,22,23,24], in line with its well-documented anorexigenic effects [25]. Christensen et al. reported a 16–39% reduction in total caloric intake among GLP-1RA-treated cohorts [16], **drawing on a narrative synthesis of multiple small dietary-intake studies involving adults with overweight or obesity. Although descriptive in nature, this review pooled findings from trials using validated 24 h dietary recalls and appetite questionnaires, providing an overarching estimate of caloric reduction across different GLP-1RA drugs**. A recent systematic review described cases of severe caloric restriction (<800 kcal/day), frequently accompanied by suboptimal intake of macro- and micronutrients, representing a clinically significant risk [22]. Despite these reductions, energy intake often remains above recommended thresholds [23,24]. Similarly, Silver et al.(2023) demonstrated a 30% reduction in total energy intake with liraglutide compared with caloric restriction alone, confirming that pharmacological appetite suppression exerts additional effects on satiety [18]. **Unlike Christensen et al., Silver et al. conducted a rigorously controlled 14-week randomised trial (n = 113) in adults with obesity and prediabetes, using 3-day weighed food diaries analysed by trained dietitians. This methodology provides a more precise estimate of nutrient intake and enables clearer observed reductions to liraglutide rather than other possible behavioural confounders.**

Protein intake shows heterogeneous trends. Some studies demonstrate a higher proportion of caloric intake from protein following GLP-1RA initiation [18], whereas others report no significant difference [23,24]. Cross-sectional data indicate, however, that most patients fail to achieve protein intake levels required to preserve lean body mass (1.2–2.0 mg/kg/day) [17]. In a subset of patients, Butsch et al. reported sarcopenia associated with GLP-1RA initiation, potentially linked to inadequate protein intake [20]. **This retrospective cohort study analysed electronic health records from adults with type 2 diabetes prescribed GLP-1RAs (n > 4000), incorporating objective biochemical markers, ICD-10 coding, and body-composition-linked surrogate measures. Unlike the intake-based methodologies of Christensen et al. and Silver et al., Butsch et al. used laboratory-verified nutritional deficiencies and clinically coded muscle-loss phenotypes, thereby providing biochemical and functional evidence of protein inadequacy rather than self-reported dietary data.**

### 3.2. Vitamin Deficiencies Observed in GLP-1 Therapy

Delayed gastric emptying, appetite suppression, and altered gut microbiota induced by GLP-1RAs may influence micronutrient absorption. A large retrospective study of 461,328 adults initiating GLP-1RA therapy found that over 22% developed at least one nutritional deficiency within 12 months, most commonly vitamin D (13.6% at 12 months), followed by thiamine, other B vitamins, and anaemia [20].

Dietary intake studies corroborate these findings, showing that GLP-1RA users frequently fail to meet Recommended Dietary Intakes for vitamins A, C, D, E, and K, as well as dietary fibre [17,19]. Obesity itself is associated with deficiencies in vitamins A, D3, E, B12, thiamine, folate, and other B vitamins [7,26]. The combined effect of obesity and GLP-1RA therapy, therefore, represents a clinically significant risk. This is particularly relevant for patients concurrently receiving metformin, which may potentiate vitamin B12 deficiency [27].

### 3.3. Mineral Deficiencies Observed in GLP-1 Therapy

GLP-1RA therapy has also been implicated in mineral deficiencies. Individuals with obesity often have inadequate intake of essential minerals such as iron, calcium, magnesium, zinc, and copper, increasing their vulnerability during GLP-1RA treatment [7]. Johnson et al. demonstrated that dietary intake of calcium, magnesium, potassium, and iron frequently fell below Dietary Reference Intakes (DRIs), while copper, phosphorus, selenium, and zinc were generally sufficient, and sodium was consumed in excess [17]. Similarly, it has been reported that average reductions of 14% in magnesium and 10% in iron intake following liraglutide treatment compared with dietary restriction alone [17].

In contrast, a randomised controlled trial found iron intake to be sufficient but identified persistent inadequacy in potassium intake across a large proportion of participants [19]. A study of long-term weight loss maintainers achieved through dietary intervention, rather than pharmacotherapy, reported higher adherence to mineral requirements compared with weight-stable individuals living with obesity [28]. These findings do, however, seem counterintuitive, and generally, we expect patients on GLP-1 RAs to experience nutrient deficiencies similar to MBS if additional supplements are not recommended. Though the overall effect may be less severe than some MBS procedures (gastric bypasses, for example), the cumulative effect of these deficiencies over time is likely to lead to serious nutritional issues.

### 3.4. Micronutrient Monitoring Protocols from MBS Guidelines

MBS is a well-established risk factor for macro and micronutrient deficiencies [29]. Accordingly, both UK and international guidelines recommend structured, lifelong monitoring of micronutrient status in patients undergoing these procedures. The American Society for Metabolic and Bariatric Surgery (ASMBS) similarly endorses pre-operative nutritional assessment and post-operative screening of iron, B12, folate, and vitamin D, with broader testing considered for patients undergoing hypoabsorptive procedures [8]. The British Obesity and Metabolic Specialist Society (BOMSS) advises comprehensive pre-operative nutritional assessment by a trained dietitian, including biochemical evaluation of ferritin, folate, vitamin B12, 25-hydroxyvitamin D, and calcium, with correction of any deficiencies before surgery [8].

Post-operative protocols outlined by BOMSS recommend regular monitoring of these micronutrients after MBS with additional evaluation of zinc, copper, and vitamin A, depending on the surgical procedure [30]. Both BOMSS and ASMBS recommend annual monitoring of vitamin B12 and vitamin D, alongside routine multivitamin supplementation containing iron, selenium, zinc, copper, and thiamine for all patients [8,30]. The European Association for Endoscopic Surgery and the Enhanced Recovery After Surgery Society also support routine nutritional surveillance, although without detailed protocols [31,32]. A summary of key bariatric surgery guidelines outlining recommended micronutrient monitoring and supplementation schedules is presented in Table 1:

### 3.5. Comparative Analysis of Nutritional Risks Between GLP-1 Therapy and MBS

Both GLP-1 receptor agonist therapy and MBS achieve substantial and sustained weight loss through appetite regulation and gastrointestinal modification; however, their nutritional consequences differ somewhat in both aetiology and magnitude. MBS, particularly diversionary procedures such as One Anastomosis Gastric Bypass or Roux-en-Y Gastric Bypass, are consistently associated with deficiencies in iron, vitamin B12, folate, calcium, and fat-soluble vitamins due to altered intake and absorption [33,34,35]. Reduced protein intake is also well documented [36,37,38] and, if uncorrected, may contribute to undesirable reductions in fat-free mass [36]. In both bariatric surgery and GLP-1RA contexts, changes in body composition reflect alterations in fat mass (FM) and skeletal muscle mass (SM), the two principal components of weight loss. Distinguishing between FM loss and SM loss is clinically relevant, as disproportionate reductions in SM are associated with sarcopenia and adverse metabolic outcomes. However, most GLP-1RA studies do not report FM and SM separately, limiting the ability to assess muscle preservation or identify subgroups at higher nutritional risk for further analysis. In contrast, the nutritional effects of GLP-1RAs are primarily due to reduced intake. Consequently, their nutritional risks cannot be assumed to mirror those following MBS.

While nutritional complications after MBS are well characterised, with comprehensive, evidence-based monitoring and supplementation protocols firmly established [8,30,31,32,39], comparable frameworks for GLP-1RA therapy do not exist. The Obesity Society and others have outlined broad nutritional priorities for patients undergoing obesity management [40], emphasising dietary counselling and surveillance; however, these recommendations lack the specificity of post-bariatric guidelines. The recent joint advisory highlights these themes by outlining practical nutritional and lifestyle priorities for patients initiating GLP-1RA therapy, including baseline assessment of dietary habits, muscle strength, body composition, and social determinants of health, as well as strategies to manage gastrointestinal side effects, preserve muscle and bone mass, and prevent micronutrient deficiencies [40]. However, these recommendations lack the structured biochemical surveillance found in established bariatric surgery pathways. Therefore, the conclusions of our review hope to extend this advisory by identifying specific nutrient vulnerabilities reported in GLP-1RA studies and by highlighting the value of adapting elements of post-bariatric monitoring frameworks to pharmacological obesity care.

More recently, it has been emphasised that GLP-1 therapy poses unique nutritional challenges, such as reduced food volume, altered feeding behaviour, and micronutrient dilution effects, and the review had called for integrated dietary intervention and biomarker monitoring protocols [41].

Given the increasing global use of GLP-1RAs and the limited duration and scope of existing studies, rigorous longitudinal research is warranted to delineate their nutritional consequences and to inform structured, evidence-based monitoring and supplementation strategies [16,40].

## 4. Discussion

This review shows that while GLP-1 receptor agonist (GLP-1RA) therapy and MBS both achieve meaningful and sustained weight loss, their nutritional implications arise through somewhat different mechanisms. In contrast to MBS, which can lead to nutritional deficiencies through a combination of reduced intake and impaired absorption (with diversionary procedures), GLP-1 RAs typically lead to nutritional deficiencies through reduced intake alone.

Despite these differences, several similarities exist in their nutritional consequences. Evidence indicates that GLP-1RA use is often associated with reduced protein intake and, in some cases, measurable loss of lean body mass. Observational studies have also reported deficiencies in vitamins D, B12, and thiamine, alongside lower intakes of minerals such as calcium, iron, and magnesium. While these abnormalities are generally less pronounced than those following some MBS procedures, they may still carry clinical significance, as the effect is likely to be cumulative with the increasing duration of their usage.

Unlike MBS, for which detailed monitoring and supplementation protocols have been developed by organisations such as BOMSS, ASMBS, and ESPEN, there are currently no standardised guidelines for the nutritional management of patients on GLP-1RAs. Adapting elements from these surgical frameworks could offer a pragmatic starting point. In our opinion, guidelines for nutritional monitoring and supplementation for Sleeve Gastrectomy (SG) can serve as a very useful starting point for patients on GLP-1 RAs. The magnitude of weight loss seen with these drugs is increasingly similar to what we see with SG, and also, there is no gastrointestinal diversion with Sleeve Gastrectomy, perhaps suggesting that most of the nutritional effects seen with this procedure are related to reduced intake rather than absorption.

Baseline screening for key deficiencies, early correction of low vitamin or mineral levels, routine supplementation strategies similar to SG protocols that exist already, and periodic reassessment during treatment may help prevent avoidable complications. However, further work is needed to identify which nutrients warrant regular monitoring and whether the supplementation doses used for those undergoing SG or BMS are appropriate for those receiving pharmacological therapy.

The emerging use of GLP-1RAs as adjuncts to bariatric surgery also warrants consideration. A recent large cohort study published in 2025 compared outcomes between more than 30,000 patients treated with MBS or GLP-1RAs and demonstrated that MBS produced substantially greater weight loss over two years than pharmacotherapy alone, with a mean total weight loss of 28.3% in the MBS group compared with 10.3% in the GLP-1RA group [12]. Although the study focused primarily on weight and cost outcomes, it underscores that combined or staged therapeutic approaches may become more common in practice. From a nutritional standpoint, such combinations could potentiate risk, as MBS and GLP-1RAs both reduce energy intake and, in bypass procedures, impair nutrient absorption. These emerging models emphasise the need for enhanced routine micronutrient monitoring when GLP-1RAs are used alongside bariatric procedures.

This review is limited by the heterogeneity of existing published evidence, as available studies can be short-term, observational, or based on dietary recall, which can be affected by reporting bias and may underestimate subclinical deficiencies. Moreover, nutritional outcomes are rarely prespecified endpoints in GLP-1RA trials, limiting causal interpretation.

Future research should focus on longitudinal studies that track changes in nutritional values over time, determine the incidence and severity of deficiencies, and evaluate the clinical benefits of supplementation. Incorporating nutritional outcomes into ongoing GLP-1 therapy trials would provide valuable evidence on whether these biochemical changes translate into measurable health effects such as anaemia, neuropathy, or bone demineralisation. In addition, developing an evidence-based framework for nutritional monitoring will be essential to support the safe, long-term use of GLP-1 therapy in obesity management.

## 5. Conclusions

GLP-1 receptor agonists have redefined obesity management, offering substantial and sustained weight loss through pharmacological mechanisms. However, emerging evidence suggests that macro and micronutrient deficiencies may occur with these drugs too, likely driven by lower dietary intake rather than impaired absorption.

Drawing lessons from MBS, incorporating baseline nutritional screening, dietetic counselling, supplementation, and periodic monitoring could help mitigate these risks. As the therapeutic use of GLP-1 agonists expands, developing evidence-based frameworks for nutritional surveillance, supplementation, and monitoring will be essential to ensure that metabolic benefits are achieved without compromising long-term nutritional health. 

## Figures and Tables

**Figure 1 nutrients-17-03659-f001:**
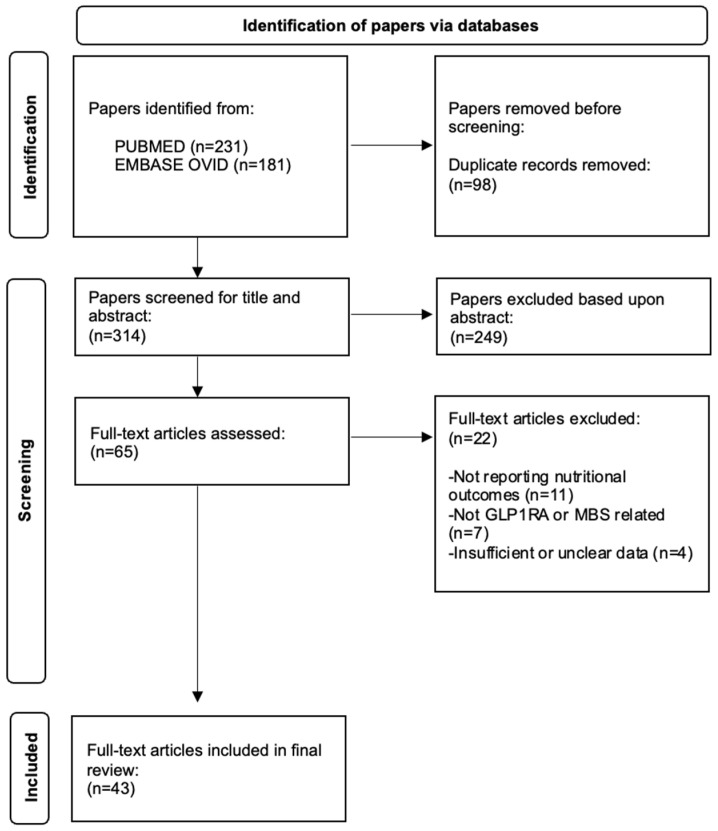
PRISMA flow diagram of study selection for the review.

**Table 1 nutrients-17-03659-t001:** Summary of selected MBS guidelines relevant to nutritional monitoring, highlighting the absence of equivalent frameworks for GLP-1 receptor agonist therapy.

Guideline	Micronutrients Monitored	Frequency of Testing	Supplementation Recommendations
ASMBS (USA, 2017) [9]	Iron, B12, folate, vitamin D, calcium	Baseline and annually	Multivitamin ± procedure-specific additions
BOMSS (UK, 2020) [8]	Ferritin, folate, B12, 25-hydroxyvitamin D, calcium, zinc, copper	Pre-op; 3, 6, 12 months; annually thereafter	Daily multivitamin + targeted repletion based on procedure
ESPEN (2023) [10]	Full micronutrient panel (no fixed list)	Annual	Case-by-case supplementation per deficiency
ERAS (2021) [11]	Emphasises early post-operative nutritional recovery	Variable	Encourages multidisciplinary dietetic follow-up rather than fixed micronutrient schedules

## Data Availability

No new data were created or analyzed in this study. Data sharing is not applicable to this article.

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
