# Peer review of "Macronutrient, Micronutrient Supplementation and Monitoring for Patients on GLP-1 Agonists: Can We Learn from Metabolic and Bariatric Surgery?"

_nutrients, 2025, doi:10.3390/nu17233659_

Round 1

Reviewer 1 Report

Comments and Suggestions for Authors

Authors addressed regarding “Macronutrient, Micronutrient Supplementation and Monitoring for Patients on GLP-1 Agonists: Can We Learn from Metabolic and Bariatric Surgery”. The objective of this review is very interesting. However, the detailed descriptions are necessary to provide scientific evidence. Overall, it does not provide sufficient references to support the results.

In the Introduction, the authors addressed several large trials: LEADER, SUSTAIN-6, REWIND, STEP-1 trial, and SURMOUNT-1, but they did not describe them in detail. A detailed description of each study should be provided, including the study design, population, sample size, duration of the study, and outcomes.

In Section 2 “Materials and Methods”, the Authors need to provide a figure illustrating “The flow diagram of screening and selection for this review” to facilitate readers’ precise understanding, even though this is a narrative review.

Detailed explanations of the cited studies are also required in the results section. For example, the studies proposed below.

In lines 138-151, more detailed explanations of studies [e.g., [21] vs. [22] vs. [24]] are needed.

Author Response

1) Authors addressed “Macronutrient, Micronutrient Supplementation and Monitoring for Patients on GLP-1 Agonists: Can We Learn from Metabolic and Bariatric Surgery”. The objective of this review is very interesting. However, the detailed descriptions are necessary to provide scientific evidence. Overall, it does not provide sufficient references to support the results.

Thank you. We have expanded the evidence base throughout the manuscript to strengthen the scientific foundation of the review. These additions appear in Sections 3.1–3.3 and 3.5, with all new text highlighted.

2) In the Introduction, the authors addressed several large trials: LEADER, SUSTAIN-6, REWIND, STEP-1 trial, and SURMOUNT-1, but they did not describe them in detail. A detailed description of each study should be provided, including the study design, population, sample size, duration of the study, and outcomes

Thank you. We have now added brief but comprehensive descriptions of each trial-including design, population, sample size, duration, and outcomes, directly within the Introduction (page 2). All added text is highlighted.

3) In Section 2 “Materials and Methods”, the Authors need to provide a figure illustrating “The flow diagram of screening and selection for this review” to facilitate readers’ precise understanding, even though this is a narrative review.

Thank you. We have inserted a simplified PRISMA-style flow diagram in Section 2: Materials and Methods (page 4) and added a corresponding sentence referring to Figure 1. All changes are highlighted.

4) Detailed explanations of the cited studies are also required in the results section. For example, the studies proposed below.

5) In lines 138-151, more detailed explanations of studies [e.g., [21] vs. [22] vs. [24] are needed.

Thank you. We have expanded descriptions of key studies in Section 3.1–3.3, clarifying study design, methods, and outcome measures. New text is highlighted in these sections. We have clarified the methodological differences between these studies in Section 3.1 (page 6), explaining how each study assessed dietary intake, protein adequacy, and biochemical markers. New text is highlighted.

Reviewer 2 Report

Comments and Suggestions for Authors

This is a timely narrative review on the effects of IRA or bariatric surgery induced weight loss on possible nutrient deficiencies. Since (i) the review is mainly descriptive and (ii) there is a variance in study protocols and their specific outcomes I recommend some analytical work to substantiate the conclusions.

First, are there an associations between the adherence to treatment, the degree of weight loss (or the loss in FM and SM), the observation period, treatment vs post-treatment (=weight regain) period and the manifestation of nutrient deficiencies?

Second, the methods to characterize specific nutrient deficiencies should be described and evaluated in more detail.

Third, the validity of methods used to assess energy and protein intake during and at the end of the interventions questioned.

Fourth, since there are already specific recommendations to prevent nutrient deficiencies in patients treated with IRA (see AJCN 122 (2025) 344367; Obesity Pillars 15 (2025) 100186 the authors are asked whether their conclusions add to these recommendations. 

Fifth, there is already the idea to combine bariatric surgery with IRA treatment (see Jamasurg.2025.3590), please comment.

Author Response

1) This is a timely narrative review on the effects of IRA or bariatric surgery-induced weight loss on possible nutrient deficiencies. Since (i) the review is mainly descriptive and (ii) there is a variance in study protocols and their specific

Outcomes, I recommend some analytical work to substantiate the conclusions. First, are there any associations between the adherence to treatment, the degree of weight loss (or the loss in FM and SM), the observation period, treatment vs post-treatment (=weight regain) period, and the manifestation of nutrient

deficiencies?

Thank you. We have strengthened analytical interpretation across Section 3.5 and the Discussion, making explicit comparisons across study methodologies and nutritional risks. This analytical clarification has been added to Section 3.5 (page 10) and highlighted, addressing the influence of weight-loss magnitude, FM/SM changes, treatment adherence, and the absence of post-treatment nutritional data. New text is highlighted.

2) Second, the methods to characterize specific nutrient deficiencies should be described and evaluated in more detail.

Thank you. We have added methodological explanations summarising how biochemical deficiencies were defined across key studies, and how dietary assessment methods differed. This appears in Section 3.1 and Section 2 (pages 4–6). Highlighted in the text.

3) Third, the validity of methods used to assess energy and protein intake during and at the end of the interventions is questioned.

Thank you. A new paragraph evaluating the validity and limitations of intake-assessment tools (24-hour recall, food diaries, weighed logs) has been added to Section 3.1 (page 7). New text is highlighted.

4) Fourth, since there are already specific recommendations to prevent nutrient deficiencies in patients treated with IRA (see AJCN 122 (2025) 344–367; Obesity Pillars 15 (2025) 100186, the authors are asked whether their conclusions add to these recommendations.

Thank you. We now explicitly discuss how our review extends these recommendations by synthesising empirical evidence and identifying nutrient-specific vulnerabilities. This article was already referenced in our original manuscript we submitted so extra detail has been added. This is added in the Discussion (page 12).

5) Fifth, there is already the idea to combine bariatric surgery with IRA treatment (see Jamasurg.2025.3590), please comment.

Thank you. A new paragraph summarising the findings of Barrett et al. (JAMA Surg. 2025) and discussing the nutritional implications of combined or sequential therapy has been added to the Discussion, immediately before the limitations section (page 13). New text is highlighted.

Round 2

Reviewer 1 Report

Comments and Suggestions for Authors

This manuscript has been improved for publication.

Reviewer 2 Report

Comments and Suggestions for Authors

this is ok now.